# Triple Semi-Circular Canal Occlusion and Cochlear Implantation: A Treatment Option for Single-Sided Menière’s Disease with Functional Deafness—A Case Series

**DOI:** 10.3390/jcm12175500

**Published:** 2023-08-24

**Authors:** Ronny Jacob, Gina Lauer, Arneborg Ernst, Rainer Seidl, Lenneke Kiefer, Philipp Mittmann

**Affiliations:** Department of Otolaryngology, Head and Neck Surgery, BG Klinikum Unfallkrankenhaus, 12683 Berlin, Germany

**Keywords:** cochlear implant, semi-circular canal occlusion, Menière’s disease

## Abstract

The surgical options for patients with single-sided Menière’s disease and functional deafness are challenging. Our case series reports the outcomes of surgical treatments of patients with single-sided Menière’s disease and functional deafness. These patients have undergone a one-staged occlusion of all semi-circular canals and cochlear implantation. Five patients (four female and one male; 62 ± 8.2 years with a range from 50 to 72 years) with single-sided Menière’s disease and functional deafness were included in this study. In all cases, the patients suffered from frequent rotational vertigo episodes for many years. Other treatment options (e.g., medication) had not yet been successful. Preoperatively, the Dizziness Handicap Inventory (DHI) of all patients indicated severe emotional, physical, and functional deficits. Patients showed a functional (near-total) deafness of the affected ear in all cases. All patients were supplied with cochlear implants in combination with a triple occlusion of all semi-circular canals in a one-stage procedure. After a short period of increased dizziness following surgery and after the activation of the cochlear implant and CI rehabilitation (auditory-verbal therapy), vertigo control and an adequate audiological outcome were achieved. The DHI showed a constant decrease after surgery. The combination of a triple semi-circular canal occlusion and cochlear implantation can be an efficient treatment for patients with single-sided Menière’s disease.

## 1. Introduction

Menière’s disease is characterized by recurrent rotational vertigo episodes, tinnitus, and sensorineural hearing loss. The baseline therapy in Europe is oral medication with betahistine or other medication (e.g., calcium antagonists) [1]. In progressive Menière’s disease, with conservative treatment, intratympanic gentamicin or intratympanic corticosteroids application can be an option. To achieve both hearing preservation and vertigo control, the best pharmacologic treatment option among the interventions compared may be an IT steroid plus high-dose betahistine, considering that IT gentamicin was found to be associated with benefits toward vertigo control but with potentially detrimental effects on hearing preservation [2,3]. Nevertheless, intratympanic medication cannot always reach an adequate control of the vertigo symptoms [1]. Even more invasive procedures (e.g., endolymphatic sac surgery) cannot safely control the vertiginous episodes in all cases. If those patients additionally suffer from a profound sensorineural hearing loss, the combination of a cochlear implant and a labyrinthectomy or occlusion of the semi-circular canals should be considered [4,5]. Subjects with end-stage Menière’s disease, and consequently unilateral hearing loss, suffer from the typical audiological disorders. Decreased sound localization, reduced speech perception abilities, and reduced quality of life due to the loss of binaural hearing can significantly impact the daily routine in patients.

Cochlear implantation (CI) is a worldwide-accepted surgical procedure for an increasing number of patients with severe-to-profound hearing loss. The criteria for cochlear implantation in Germany have expanded over the past decades, nowadays including recipients with unilateral aided residual hearing up to 60% of speech understanding at 65 dB SPL. The plugging of the semi-circular canals can be performed via a transmastoidal approach or via the middle fossa approach for the superior canal. The approach is established and well known from semi-circular dehiscence surgery. Surgery can be performed with only minimal trauma and even without affecting the hearing threshold. Early feasibility studies on hearing preservation after canal plugging were first observed in guinea pigs and later proved in humans. Kontorinis et al. showed that 11 out of 30 patients that were enrolled had significantly improved DHI scores, while hearing did not change significantly. They concluded that the transmastoid plugging of the superior semi-circular canal can safely and significantly improve the vestibular symptoms of patients with SCDS, as well as the auditory symptoms in a substantial number of patients in a hearing-preservation way [6]. The study by the House group showed similar results in a group of 24 ears. Plugging the semi-circular canal is a safe method with vestibular symptom improvement in 35.7% of patients, and word recognition scores did not significantly change postoperatively [7].

Combining vestibular surgery and cochlear implantation is quite a new approach to the neurotological society. Saber et al. performed a single-patient procedure that involved plugging the posterior semi-circular canal and performing cochlear implantation in the ipsilateral ear. The patient had benign paroxysmal positional vertigo with single-sided deafness. The outcome of the procedure was overall successful in terms of vertigo and auditory rehabilitation, but the tinnitus worsened after surgery [8].

While a labyrinthectomy can efficiently control the episodes of vertigo, the procedure also results in complete vestibular deafferentation [9]. In a single-center retrospective study between 2003 and 2019, seventy-two patients underwent unilateral labyrinthectomy. Indication criteria were mostly drop attacks or failure of treatment with intratympanic gentamicin [10]. The mean preoperative word recognition score was 36.4% in the affected ear. The Gruppo Otologico investigated the charts of 22 patients undergoing labyrinthectomy and cochlear implantation in the same ear for intractable vertigo and hearing loss [11]. A total of 67% of the patients had complete resolution of the vestibular symptoms in their operated ear, but speech audiometry after cochlear implantation was not significantly better [11]. They concluded that in Menière’s disease with vertigo and severe hearing loss, labyrinthectomy and cochlear implantation can be a reasonable solution, but in the elderly, more failures and postoperative instability are observed.

However, a less destructive approach (canal occlusion) can be particularly helpful for patients who develop Menière’s disease in the contralateral ear or for the elderly with an age-related impairment of the balance system. Experimental evidence indicates that a triple semi-circular canal occlusion could be an effective option for controlling episodes of rotational vertigo in Menière’s disease [12]. In a small cohort of three patients, Gill et al. performed a triple semi-circular canal occlusion for the treatment of Menière’s disease. Two patients showed no effect on hearing whereas one patient suffered a unilateral 30 dB hearing loss. Vertigo control was excellent in two patients. The aim of our study was to investigate vertigo control and hearing performance after a triple semi-circular canal occlusion and simultaneous cochlear implantation in patients with unilateral Menière’s disease.

## 2. Materials and Methods

After unsuccessful conservative treatment to control the episodes of vertigo, one male and four female patients with single-sided Menière’s disease [13] underwent surgical treatment. Inclusion criteria were single-sided Menière’s Disease, age over 18 years, oral medication for over a year, and severe hearing loss in the affected ear. The mean age was 62 ± 8.2 years with a range from 50 to 72 years. All patients were on medication (from 3 to 19 years) and reported frequent episodes of rotational vertigo as the most impairing symptom (2–3× week on average). The treatment with gentamicin (intratympanic application) or other procedures was declined by the patients. Exclusion criteria included patients younger than 18 years of age, a previous operation in the affected ear, and gentamicin treatment before surgery.

The patients were treated between August 2013 and October 2015 using the identical surgical technique of one-stage cochlear implantation and the occlusion of all three ipsilateral semi-circular canals. For cochlear implantation, a posterior tympanotomy and a round window approach were chosen [14]. Occlusion of the three semi-circular canals was performed before the implantation by reducing the drill speed to 10,000/min. The labyrinth was skeletonized to identify all semi-circular canals. The superior bony layer of the semi-circular canal was drilled down with a diamond burr (2.3 mm) until the endolymphatic duct shone through the last bony layer (blue lining) (Figure 1 and Figure 2). Leaving the endolymphatic duct intact, the temporalis fascia was pushed down in the canal. After sealing with fascia, the canal was filled with bone wax (Figure 3). The area of each canal was covered afterward with a muscle patch and bone pate. Finally, the sealing patch was secured with fibrin glue (Figure 4) [5,15].

The subjective and objective vestibular findings were assessed preoperatively, directly postoperatively, as well as 6–8 weeks and 6 months after surgery. For this purpose, the Dizziness Handicap Inventory (DHI) with its subscales (physical (DHIp), functional (DHIf), and emotional (DHIe)) was used as a reliable and validated instrument for the functional outcome–evaluation [16]. Patients filled in the DHI by themselves. The questionnaire was handed to each patient, who read and filled it by themselves.

Preoperatively, the video head impulse test was performed in two patients (normal responses for all semi-circular canals). Cervical-evoked vestibular myogenic potentials (cVEMPs) were recorded in all patients but could only be elicited in two patients. None of the patients showed spontaneous nystagmus. Caloric tests were not performed.

The audiological results of the cochlear implantation were assessed by testing numbers and monosyllabic words (Freiburger test) (Table 1). The study was approved by the institutional review board at the BG Klinikum Unfallkrankenhaus Berlin (ukb-hno-2013/4). The study was conducted according to the principles expressed in the Declaration of Helsinki. Statistical evaluation was performed using SPSS (Version 22.0; IBM Co., Armonk, NY, USA). For the comparison between the different DHI scores, a two-way repeated measures ANOVA was run.

## 3. Results

Before the surgical treatment was started, the DHI suggested a severe functional, emotional, and physical impairment of all included individuals. Preoperatively, the mean DHI was quantified as serious, regarding the functional (DHIf 20.8 ± 3.0), and emotional (DHIe 22.4 ± 6.5) as well as moderate, concerning the physical (DHIp 10 ± 6.2) deficits (Table 2).

All patients reported a complete cessation of Menière’s typical vertigo episodes, but one female patient postoperatively described unstableness instead of the pre-existing vertigo episodes. After revision surgery (labyrinthectomy after three months), she became free of vertigo episodes. The data of this patient included for statistical analysis were those after revision surgery.

Three of the five patients reported an immediate improvement in symptoms (vertigo episodes and unstableness) after surgery. The postoperative DHI of these three patients already indicated a statistically significant decrease in the functional, physical, and emotional subscales during the postoperative hospital stay.

After 6 weeks of compensation, four subjects (and one subject after labyrinthectomy) were almost free of symptoms. Slight vertigo or feelings of insecurity only occurred with challenging activities like sports or hard physical work, as described by some of the patients. Ordinary everyday activities could be performed without any limitations. Six months after the surgery, this status even improved (Figure 5). All subjects did not undergo a specific vestibular rehabilitation program.

A two-way repeated measures ANOVA was run to monitor the effects over time for DHIp, DHIf, and DHIe. For the DHIp, there was a statistically significant difference in DHIp scores between time points, F(3, 12) = 25.580, *p* < 0.0005. Post hoc analysis with a Bonferroni adjustment revealed that DHIp scores were statistically significantly decreased from pre-intervention to six weeks (14.2 (95% CI, 1.238 to 27.162) *p* = 0.036) and from pre-intervention to six months (19.4 (95% CI, 6.511 to 32.289) *p* = 0.011).

For the DHIf scores, there was a statistically significant difference in DHIf scores between time points, F(1.417, 5.667) = 17.134, *p* = 0.005. Post hoc analysis with a Bonferroni adjustment revealed that DHIf scores were statistically significantly decreased from pre-intervention to six weeks (10.4 (95% CI, 2.638 to 18.162) *p* = 0.017) and from pre-intervention to six months (18.2 (95% CI, 14.319 to 22.081) *p* < 0.005).

For the DHIe scores, there was a statistically significant difference in DHIe scores between time points, F(3, 12) = 22.580, *p* < 0.0005. Post hoc analysis with a Bonferroni adjustment revealed that DHIe scores were statistically significantly decreased from pre-intervention to six weeks (14.2 (95% CI, 1.238 to 27.162) *p* = 0.036) and from pre-intervention to six months (19.4 (95% CI, 6.511 to 32.289) *p* = 0.011).

All patients included in this study had profound sensorineural hearing loss. After cochlear implantation and auditory rehabilitation, all patients reported satisfying speech recognition (Table 3).

## 4. Discussion

Treatment of vertigo and hearing rehabilitation in patients with Menière’s disease is challenging, particularly when the medical treatment options (incl. gentamicin) do not yield sufficient results. In cases of recurrent episodes of vertigo—an ipsilateral profound hearing loss—as well as normal or near-to-normal hearing and proper vestibular function of the contralateral side, ablative inner ear surgery should be considered. Labyrinthectomy is a well-established method for the effective treatment of vertigo episodes or Tumarkin crises caused by Menière’s disease with the side effect of cochlear deafness [17]. In a previous study, we reported the combination of labyrinthectomy and simultaneous cochlear implantation in five patients with single-sided Menière’s disease and pre-existing ipsilateral deafness [5].

Transmastoid plugging of the semi-circular canals is one surgical option in patients with vestibular symptoms caused by the canal dehiscence syndrome or benign paroxysmal vertigo, as conservative medical treatment is limited. Results from human studies show that vertigo control can be achieved via semi-circular canal plugging, and hearing is not affected [15]. Several studies showed that in 10% to 35% of patients with single-sided Menière’s disease, so-called silent contralateral endolymphatic hydrops can be detected [18,19,20,21]. Therefore, a labyrinthectomy might severely impair the overall balance and postural control in the long term, so a more selective procedure as described in this paper seems to be more favorable.

Results of unilateral labyrinthectomy vary. The Gruppo Otologico showed that 67% of the patients had complete resolution of vestibular symptoms in their operated ear. On the other hand, speech audiometry after simultaneous cochlear implantation was not significantly better postoperative [11]. They conclude that in Menière’s disease with vertigo and severe hearing loss, labyrinthectomy and cochlear implantation can be a reasonable solution, but in the elderly, more failures and postoperative instability are observed [11]. Perkins et al. focus their study on audiological results after simultaneous labyrinthectomy and cochlear implantation in unilateral Menière’s disease. Patients with unilateral Menière’s Disease who underwent simultaneous labyrinthectomy and cochlear implantation experienced improvements in sound localization, speech understanding, tinnitus severity, and quality of life with the device [22]. In contrast to the study by the Gruppo Otologico, these subjects were younger, but only three subjects were investigated. Our population was slightly younger than in the study by Sykopetrites et al. and more comparable to Perkins et al. Regarding the audiological results, our data seem to underline the data from Perkins et al.

The current data regarding semi-circular canal plugging and simultaneous cochlear implantation are limited. In a small cohort of three patients, Gill et al. performed a triple canal occlusion for the treatment of Menière’s disease. Vertigo control was excellent in two-thirds of the patients. Saber et al. showed in a single-patient procedure that the plugging of the posterior semi-circular canal and simultaneously performing cochlear implantation in the ipsilateral ear led to overall success in terms of vertigo control and auditory rehabilitation [8].

Conservative and small invasive procedures such as intratympanic glucocorticoids or gentamicin can be an option in patients with severe vertigo symptoms. Ahmadzai et al. point out that to achieve both hearing preservation and vertigo control, the best pharmacologic treatment option among the interventions compared may be intratympanic steroid plus high-dose betahistine, considering that intratympanic gentamicin was found to be associated with benefits toward vertigo control but with potentially detrimental effects on hearing preservation [2]. Similar results were found by Hao et al. They support the findings, by Ahmadzai et al., that intratympanic gentamicin and glucocorticoids both showed beneficial effects compared with placebo treatment in vertigo management of Menière’s disease [2,3]. Lee et al. also found similar results but underline that in their results, a substantial amount of heterogeneity and publication bias was found [23].

Our results underline the hypothesis by Gill et al. and Saber et al. and demonstrate that the functional outcome is similar to a labyrinthectomy with respect to the control of Menière’s episodes. The DHI showed a comparable reduction in emotional, functional, and physical impairment. Both techniques led to a highly significant reduction in the DHI to nearly normal values in the long term after a period of at least 6 months of compensation [5]. The functional outcome (DHIf) even followed a better time course in those patients of the labyrinthectomy group [5]. The DHI indicated significantly faster functional improvement after surgery as compared to labyrinthectomy [5]. However, no significant differences in the emotional and physical subscales in the time course were found when comparing both groups. The same holds true for the hearing outcome after cochlear implantation.

The DHI of four out of five patients who underwent triple semi-circular canal occlusion surgery followed a similar time course as the patients who directly underwent a labyrinthectomy. They preoperatively suffered recurrent episodes of vertigo from twice a week up to several times a day. Postoperatively, after a few weeks of vestibular compensation, these patients had a statistically significant reduction in their functional, physical, and especially, emotional subscales of the DHI to nearly normal. Just in one case, no significant improvement of vertigo could be achieved, so revision surgery (labyrinthectomy) had to be performed.

Nevertheless, a triple semi-circular canal occlusion should be chosen as an option before performing a labyrinthectomy for the reasons outlined above. In four out of the five patients in this series, the saccule and utricle (gravity receptors) could be successfully preserved.

Therefore, the least traumatic approach in canal occlusion or revision surgery should be chosen [24,25,26]. Furthermore, patients gain a benefit from the auditory rehabilitation of the hearing loss with a cochlear implant. Our results, as well as those previously published by Sader et al., strongly advocate for simultaneous cochlear implantation and triple semi-circular canal occlusion in this population due to their substantial benefits. Similar to the mentioned studies, our study is limited to a small number of participants. In addition, the surgeon and the patient are not blinded to the procedure. In all patients, no adverse events were recorded. Neither facial paresis, cerebral-spinal-fluid leak, nor wound infection or infection of the cochlear implant were recorded.

Nevertheless, regardless of the small number of patients, our results demonstrate that simultaneous triple semi-circular plugging and cochlear implantation is a successful and safe procedure in suppressing vertigo symptoms and restoring hearing in unilateral Menière’s disease.

## 5. Conclusions

For patients with single-sided Menière’s disease, functional deafness, and frequent episodes of vertigo, simultaneous cochlear implantation with the occlusion of all three semi-circular canals can be an efficient therapeutical procedure. This procedure is as efficient as the labyrinthectomy with simultaneous cochlear implantation in vertigo control. However, the technical aspects of occlusion surgery are more challenging since the preservation of the otolith organs should lead to better postural control postoperatively and would be a huge advantage in those patients who develop a bilateral Menière’s disease.

## Figures and Tables

**Figure 1 jcm-12-05500-f001:**
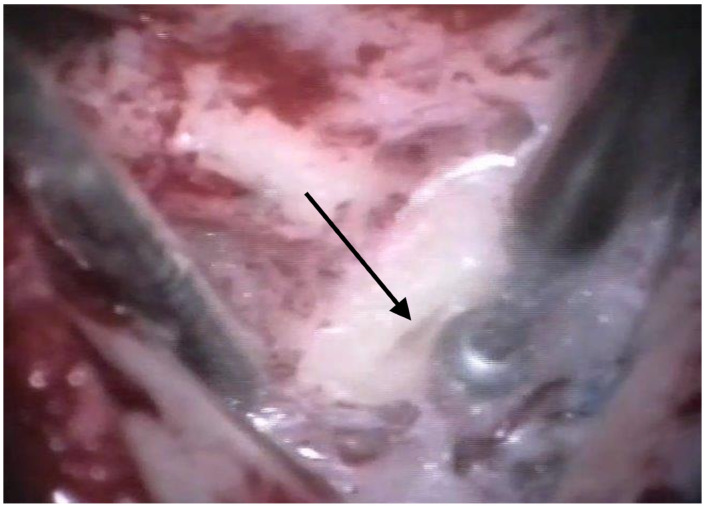
Blue lining of the lateral semi-circular canal with the diamond burr. The arrow points to the lateral semi-circular canal.

**Figure 2 jcm-12-05500-f002:**
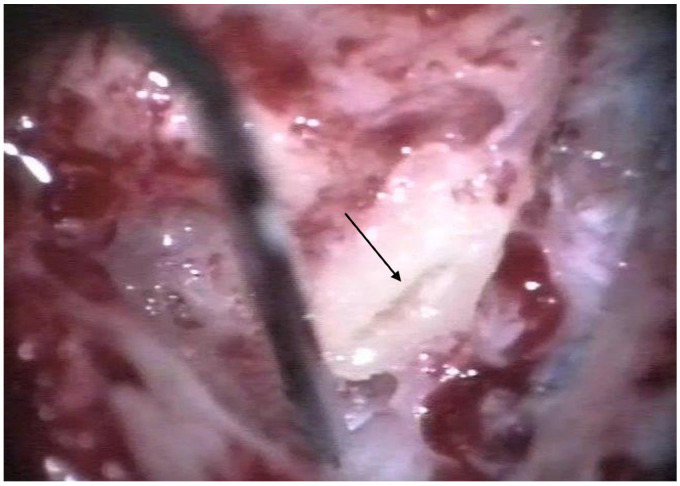
After blue lining of the lateral semi-circular canal, the canal was filled with fascia. The arrow points to the lateral semi-circular canal.

**Figure 3 jcm-12-05500-f003:**
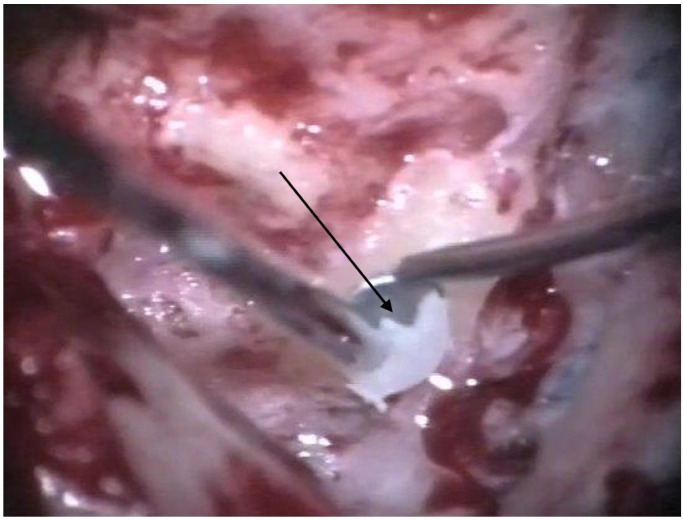
Occlusion with bone wax. The arrow points to the wax applied to the canal.

**Figure 4 jcm-12-05500-f004:**
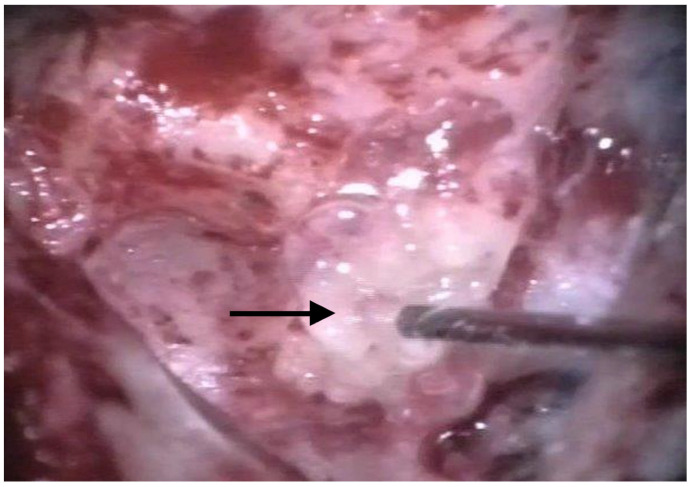
In the last step, fibrin glue was used to seal the plug. The arrow points to the fibrin glue.

**Figure 5 jcm-12-05500-f005:**
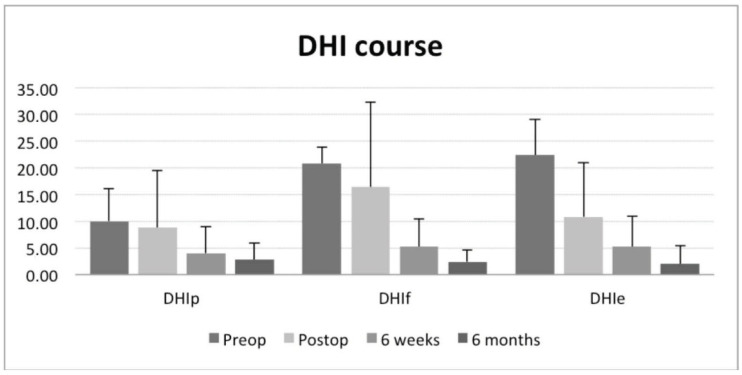
Time course of mean DHI score.

**Table 1 jcm-12-05500-t001:** Patient demographics.

Patient	Age	Sex	Episodes of Vertigo in 1 Year	Mean Pure Tone Average between 500 Hz and 4 kHz	Word Recognition Score	DHI
Patient 1	66	w	24	100 dB	0%	46
Patient 2	50	w	52	95 dB	0%	64
Patient 3	60	m	36	130 dB	0%	46
Patient 4	60	w	20	130 dB	0%	60
Patient 5	72	w	78	130 dB	0%	50

**Table 2 jcm-12-05500-t002:** Time course of mean DHI score.

Quality	Preop.	Postop.	6 Weeks	6 Months
Physical deficit	10 ± 6.2	8.8 ± 10.7	4.0 ± 4.9	2.8 ± 3.0
Functional deficit	20.8 ± 3.0	16.4 ± 15.9	5.2 ± 5.2	2.4 ± 2.2
Emotional deficit	22.4 ± 6.5	10.8 ± 10.2	5.2 ± 5.8	2.0 ± 3.5

**Table 3 jcm-12-05500-t003:** Individual speech recognition before and after cochlear implantation.

	Preop Monosyl at 65 dB	Postop Numbers at 65 dB	Postop Numbers at 45 dB	Postop Monosyl Words at 65 dB
Patient 1	0	100%	100%	55%
Patient 2	0	100%	70%	75%
Patient 3	0	100%	50%	65%
Patient 4	0	100%	80%	60%
Patient 5	0	100%	60%	25%

## Data Availability

All data are available from the corresponding author.

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
