# Peer review of "Triple Semi-Circular Canal Occlusion and Cochlear Implantation: A Treatment Option for Single-Sided Menière’s Disease with Functional Deafness—A Case Series"

_jcm, 2023, doi:10.3390/jcm12175500_

Round 1

Reviewer 1 Report

Thank you for the opportunity to review this manuscript. I have only several minor comments that should be considered:

1) The introduction lacks rationale for the current study. The rationale does not become clear until the discussion section in which the authors highlight the less invasive nature of the proposed approach compared to previous approaches. This information should be included in the introduction to help the reader see the rationale and underlying hypothesis of the current study.

2) The abstract includes several abbreviations that are not defined (ssc, DHI, CI). Also, I'd suggest getting rid of the capitalization of the word "NO" on line 18 of the abstract.

3) Lines 5 - 6 on Page 2: Please provide additional information as to why the patients declined gentamicin treatment and other treatment options. These are less invasive than the proposed treatment and, as noted in the discussion, are a typical treatment approach before moving on to more aggressive and invasive approaches.

4) Lines 32 - 34 on Page 2: It reads as if for subject B the initial surgery resolved the vertigo attacks but introduced a new symptom of general instability. A second surgery was then performed, after which "she became free of vertigo attacks." Wasn't she free from vertigo attacks after the 1st surgery and the purpose of the 2nd was to resolve the instability? Please clarify.

5) Define the abbreviations DHIf, DHIp, and DHIe.

6) Table I is never called out in the manuscript text.

7) Lines 14 - 24 on Page 4: The citation style in this paragraph is different from that used throughout the manuscript.

Author Response

1) The introduction lacks rationale for the current study. The rationale does not become clear until the discussion section in which the authors highlight the less invasive nature of the proposed approach compared to previous approaches. This information should be included in the introduction to help the reader see the rationale and underlying hypothesis of the current study.

            Reply: Thank you for the comment, please find the introduction revised and rewritten.

2) The abstract includes several abbreviations that are not defined (ssc, DHI, CI). Also, I'd suggest getting rid of the capitalization of the word "NO" on line 18 of the abstract.

            Reply: Thank you for the comment. All abbreviation were either taken out or defined first.

3) Lines 5 - 6 on Page 2: Please provide additional information as to why the patients declined gentamicin treatment and other treatment options. These are less invasive than the proposed treatment and, as noted in the discussion, are a typical treatment approach before moving on to more aggressive and invasive approaches.

            Reply:

4) Lines 32 - 34 on Page 2: It reads as if for subject B the initial surgery resolved the vertigo attacks but introduced a new symptom of general instability. A second surgery was then performed, after which "she became free of vertigo attacks." Wasn't she free from vertigo attacks after the 1st surgery and the purpose of the 2nd was to resolve the instability? Please clarify.

            Reply:

5) Define the abbreviations DHIf, DHIp, and DHIe.

            Reply: Please see above, abbreviations were defined when first mentioned.

6) Table I is never called out in the manuscript text.

            Reply: Thank you for the observation, please find this corrected in the results section.

7) Lines 14 - 24 on Page 4: The citation style in this paragraph is different from that used throughout the manuscript.

            Reply: Thank you for the observation, please find this corrected.

Reviewer 2 Report

The authors wrote an article about Triple Semicircular Canal occlusion and cochlear implantation as A treatment option for single- sided Menière's Disease with  functional Deafness.

The article should be very interesting, but it needs a lot of corrations.

1. Please add a table about the study population, including pre-operartive DHI; Tinnitus handicap inventory THI; PTA; word recognition score; number of vertigo crisis in 1 year; a VAS about headache and fullness.

2. The statistical analysis is not performed, please include it in the methods.

3. The most interesting part is the 3 canals obliteration; please insert figures or video with the most important step of surgery

Author Response

  1. Please add a table about the study population, including pre-operartive DHI; Tinnitus handicap inventory THI; PTA; word recognition score; number of vertigo crisis in 1 year; a VAS about headache and fullness.

Reply: A table was added to the manuscript. We agree that the suggested scales would be interesting and will be part of following studies. Unfortunately, THI and VAS were not part of the protocol and cannot be presented.

  1. The statistical analysis is not performed, please include it in the methods.

            Reply: Please find this added in the M&M and results section.

  1. The most interesting part is the 3 canals obliteration; please insert figures or video with the most important step of surgery.

            Reply: Please find the intraop photos in the manuscript.

Reviewer 3 Report

Thank you for the opportunity to review this study, which presents an interesting idea on the topic, my considerations are detailed below:

Title:

- What the type of study? Pilot study? Case series? It needs to be in the title.

Abstract:

- I suggest to the authors, for a better understanding of the readers, the inclusion of the subtopics in the abstract (introduction, objectives, methods, results and conclusion)

- I suggest changing the word attack, by episodes or history of rotational vertigo

- The authors do not mention what they used in the methods, however, in the results they already mention DHI results, which makes the abstract confusing for those who are reading it.

- The Abstract is very confusing, the type of study is not clear, the wording is not clear on how each stage was conducted, the abstract needs to be rewritten.

- I suggest authors rewrite the abstract, certainly the inclusion of subtopics will help authors to make the text more understandable and readers to better understand how the stages of this study were carried out.

Introduction:

- I suggest changing the word attack, by episodes or history of rotational vertigo throughout the study

- What are the advantages of triple occlusion of SSC for other types of treatments?

- Are there any adverse effects of these interventions? Which?

- Authors should demonstrate in the introduction the advantages and disadvantages of treatment methods for vertigo and why this type of approach studied in this article should be more indicated.

- Where is the purpose of the study?

Methods:

- What type of study?

- How were the patients recruited? Where were the patients recruited from?

- What are the inclusion criteria?

- What are the exclusion criteria?

- The study appears to be a pilot study, therefore, it needs approval from the ethics committee and proof of registration of the protocol, published on an international platform, such as Clinical trials, for example. Please mention the last aspect in the article.

- Objective and subjective vestibular findings were assessed by which instrument? This needs to be better detailed in the article.

- Who applied the DHI? Was the patient asked? Did the patient himself read and mark the questions? The application of the DHI should be better detailed in the article.

- The surgical technique should be better detailed, to promote its replication in future studies and the authors should mention references for its use.

- I suggest including photos of the procedures to improve readers' understanding of the surgery.

- The authors need to rewrite the methods in order to explain in a temporal way how each of the steps occurred in the study. For example, the authors mention the performance of surgeries and after reporting this, they mention that before the surgeries the evaluation of the function of the CCS was carried out, that is, the text goes back and forth, it does not present a chronological sequence of how everything happened in the study. Authors need to rewrite the methods in a chronological sequence of how each step happened in their study.

- I suggest including the full name of the university/institute where the local ethics committee approved the study.

- What statistical analyzes were carried out? What statistical tests were used in this study? This information must be included in the article.

Results:

- The authors mention significant differences, however, they do not mention the (p-value) of any of these analyses.

- The authors mention two groups A and B in the results, however, in the methods there is no mention of this, that is, there is no way to interpret these results.

- The authors also mention in the text of the results the abbreviations DHIp, DHIf and DHIe without any explanation of what these acronyms are, they are used in Figure 1 as well.

Discussion:

- The discussion is very poor in terms of scientific evidence.

- The authors discuss findings with definitions and do not physiologically explain why this type of surgery would be more beneficial than other treatments.

- In addition, none of the references in this article are updated, published in the last 5 years and three recent meta-analyses mention that the use of intratympanic gentamicin and glucocorticoids are effective to control vertigo episodes, diverging from what the authors mention in that study:

1- Ahmadzai et al (2020) - Pharmacologic and surgical therapies for patients with Meniere’s disease: A systematic review and network meta-analysis.

2- Lee et al (2021) - Intratympanic steroid versus gentamicin for treatment of refractory Meniere's disease: A meta-analysis.

3- Hao et al (2022) - Effects of intratympanic gentamicin and intratympanic glucocorticoids in Ménière’s disease: a network meta-analysis.

- The authors did not mention the limitations of the study.

- Another absence was the presence or absence of adverse events from the surgery, not reported by the authors and is very important, as it helps the patient's clinical decision-making.

- Another important clinical aspect not mentioned was the indication for vestibular rehabilitation, as patients tend to remain with persistent dizziness and vestibular rehabilitation exercises for these patients are essential.

Author Response

Title:

- What the type of study? Pilot study? Case series? It needs to be in the title.

            Reply: Thank you for the comment, please find this attached to the title.

Abstract:

- I suggest to the authors, for a better understanding of the readers, the inclusion of the subtopics in the abstract (introduction, objectives, methods, results and conclusion)

Reply: Thank you for the comment, we agree that it would give the abstract a structure. Looking in the Instructions for Authors on the JCM webside, headings should be kept out of the abstract: “The abstract should be a total of about 200 words maximum. The abstract should be a single paragraph and should follow the style of structured abstracts, but without headings:”

- I suggest changing the word attack, by episodes or history of rotational vertigo

            Reply: The term attack was replaced by episodes through the manuscript.

- The authors do not mention what they used in the methods, however, in the results they already mention DHI results, which makes the abstract confusing for those who are reading it.

            Reply: Thank you for the observation. Please find this mentioned in the reworded abstract.

- The Abstract is very confusing, the type of study is not clear, the wording is not clear on how each stage was conducted, the abstract needs to be rewritten.

Reply: Thank you for the comment. The abstract was reworded, however the headlines were left out due to the instructions for authors.

- I suggest authors rewrite the abstract, certainly the inclusion of subtopics will help authors to make the text more understandable and readers to better understand how the stages of this study were carried out.

            Reply: Thank you, suggestions were overtaken.

Introduction:

- I suggest changing the word attack, by episodes or history of rotational vertigo throughout the study

            Reply: Changed throughout the manuscript

- What are the advantages of triple occlusion of SSC for other types of treatments?

Reply: We present the first study with a triple occlusion of SSC with cochlear implantationin Menière’s Disease. The advantages are a less traumatic approach than labyrinthectomy or nerval dissection of the vestibular nerve. In contrast to labyrinthectomy or dissection of the vestibular nerve, the inner ear canal remains untouched and hence the risk for a liquor fistula.

- Are there any adverse effects of these interventions? Which?

Reply: Adverse events are mainly postoperative vertigo and the risk of hearing loss. Our patients all showed functional deafness before surgery and were simultaneously implanted with a cochlear implant. The risk of postoperative hearing loss is hence not applicable.

- Authors should demonstrate in the introduction the advantages and disadvantages of treatment methods for vertigo and why this type of approach studied in this article should be more indicated.

            Reply: The introduction was expanded. Pease find the new introduction in the paper.

- Where is the purpose of the study?

Reply: The purpose of the study was to show that before extended surgery (e.g. labyrinthectomy) less invasive surgery should be the first option in cases of single sided Menière. We hope this is more clearly in the new introduction.

Methods:

- What type of study?

            Reply: Pilot study

- How were the patients recruited? Where were the patients recruited from?

Reply: Patients were referred to the hospital with a history of long term MD. No active recruiting process was obtained.

- What are the inclusion criteria?

            Reply: Please find the inclusion criteria added in the M&M section.

- What are the exclusion criteria?

            Reply: Please find the exclusion criteria added in the M&M section.

- The study appears to be a pilot study, therefore, it needs approval from the ethics committee and proof of registration of the protocol, published on an international platform, such as Clinical trials, for example. Please mention the last aspect in the article.

Reply: The study was approved by the local ethic committee. Please find this mentioned in the M&M section.

- Objective and subjective vestibular findings were assessed by which instrument? This needs to be better detailed in the article.

            Reply: Thank you for the comment, vestibular testings were extended in the M&M section.

- Who applied the DHI? Was the patient asked? Did the patient himself read and mark the questions? The application of the DHI should be better detailed in the article.

Reply: The DHI was handed to the patient. Patients read and marked the questions themselves. Please find this added in the M&M section.

- The surgical technique should be better detailed, to promote its replication in future studies and the authors should mention references for its use.

            Reply: Please find a more detailed and referenced part in the M&M section.

- I suggest including photos of the procedures to improve readers' understanding of the surgery.

                        Reply: Please find photos included.

- The authors need to rewrite the methods in order to explain in a temporal way how each of the steps occurred in the study. For example, the authors mention the performance of surgeries and after reporting this, they mention that before the surgeries the evaluation of the function of the CCS was carried out, that is, the text goes back and forth, it does not present a chronological sequence of how everything happened in the study. Authors need to rewrite the methods in a chronological sequence of how each step happened in their study.

            Reply: Please find the M&M section reworded and expanded.

- I suggest including the full name of the university/institute where the local ethics committee approved the study.

            Reply: Please find the information added.

- What statistical analyzes were carried out? What statistical tests were used in this study? This information must be included in the article.

Reply: The statistical information is in the results part but is now also mentioned in the M&M section.

Results:

- The authors mention significant differences; however, they do not mention the (p-value) of any of these analyses.

            Reply: Please find the p-values on page 8 and 9 of the manuscript.

- The authors mention two groups A and B in the results, however, in the methods there is no mention of this, that is, there is no way to interpret these results.

            Reply: Thank you for the comment. These groups were taken out.

- The authors also mention in the text of the results the abbreviations DHIp, DHIf and DHIe without any explanation of what these acronyms are, they are used in Figure 1 as well.

Reply: Thank you for the observation. Please find the abbreviations now fully written out when using the first time.

Discussion:

- The discussion is very poor in terms of scientific evidence.

            Reply: Please find the discussion extended and revised in total.

- The authors discuss findings with definitions and do not physiologically explain why this type of surgery would be more beneficial than other treatments.

Reply: The physiological difference between labyrinthectomy and triple canal plugging is the preservation of saccule and utricle. Therefore, compensation can be faster and rehabilitation should be faster and more comfortable. Please find this mentioned in the discussion.

- In addition, none of the references in this article are updated, published in the last 5 years and three recent meta-analyses mention that the use of intratympanic gentamicin and glucocorticoids are effective to control vertigo episodes, diverging from what the authors mention in that study:

1- Ahmadzai et al (2020) - Pharmacologic and surgical therapies for patients with Meniere’s disease: A systematic review and network meta-analysis.

2- Lee et al (2021) - Intratympanic steroid versus gentamicin for treatment of refractory Meniere's disease: A meta-analysis.

3- Hao et al (2022) - Effects of intratympanic gentamicin and intratympanic glucocorticoids in Ménière’s disease: a network meta-analysis.

Reply: As the discussion was revised more recent studies and the above-mentioned studies were included.

- The authors did not mention the limitations of the study.

            Reply: Please find the limitations of the study included in the discussion

- Another absence was the presence or absence of adverse events from the surgery, not reported by the authors and is very important, as it helps the patient's clinical decision-making.

            Reply: Thank you for the observation. Please find this included in the discussion.

- Another important clinical aspect not mentioned was the indication for vestibular rehabilitation, as patients tend to remain with persistent dizziness and vestibular rehabilitation exercises for these patients are essential.

Reply: The reviewer is right. Vestibular rehabilitation is a key part after any neurotological procedure. Please find this mentioned in the discussion. Our patients did not participate in a specific vestibular rehabilitation over the period of six months. Please find this also mentioned in the results section.

Round 2

Reviewer 3 Report

The authors did a good job, the inclusion of the figures will guide readers to how the authors accomplished this step. I believe that the article is now suitable to be accepted for publication.